# EVM: An Educational Virtual Reality Modeling Tool; Evaluation Study with Freshman Engineering Students

**Julián Conesa-Pastor [1,\*] and Manuel Contero [2]**

1   Departamento de Estructuras, Construcción y Expresión Gráfica, Campus Muralla del Mar,
    Universidad Politécnica de Cartagena, C/Doctor Fleming s/n, 30202 Cartagena, Spain
2   I3B, Universitat Politècnica de València, Camino de Vera s/n, 46022 Valencia, Spain; mcontero@upv.es
*   Correspondence: julian.conesa@upct.es; Tel.: +34-968-326-477

**Abstract:** Educational Virtual Modeling (EVM) is a novel VR-based application for sketching and modeling in an immersive environment designed to introduce freshman engineering students to modeling concepts and reinforce their understanding of the spatial connection between an object and its 2D projections. It was built on the Unity 3D game engine and Microsoft's Mixed Reality Toolkit (MRTK). EVM was designed to support the creation of the typical parts used in exercises in basic engineering graphics courses with a special emphasis on a fast learning curve and a simple way to provide exercises and tutorials to students. To analyze the feasibility of using EVM for this purpose, a user study was conducted with 23 freshmen and sophomore engineering students that used both EVM and Trimble SketchUp to model six parts using an axonometric view as the input. Students had no previous experience in any of the two systems. Each participant went through a brief training session and was allowed to use each tool freely for 20 min. At the end of the modeling exercises with each system, the participants rated its usability by answering the System Usability Scale (SUS) questionnaire. Additionally, they filled out a questionnaire for assessment of the system functionality. The results demonstrated a very high SUS score for EVM (M = 92.93, SD = 6.15), whereas Trimble SketchUp obtained only a mean score of 76.30 (SD = 6.69). The completion time for the modeling tasks with EVM showed its suitability for regular class use, despite the fact that it usually takes longer to complete the exercises in the system than in Trimble SketchUp. There were no statistically significant differences regarding functionality assessment. At the end of the experimental session, participants were asked to express their opinion about the systems and provide suggestions for the improvement of EVM. All participants showed a preference for EVM as a potential tool to perform exercises in the engineering graphics course.

**Keywords:** virtual reality; sketching; modeling; engineering graphics

## 1. Introduction and Related Work

The application of Virtual Reality (VR) technology to 3D modeling was one of the first uses that VR pioneers gave to the primitive hardware that was available in the 1990s. 3DM [1] was one of the first systems with the capability of creating 3D geometry. The system was implemented with a VPL EyePhone head-mounted display (HMD) with a 442 × 238 point LCD screen, a field of view of 108° × 76°, and a price of around $9000 [2]. The resolution was so low that the text labels on the buttons were very difficult to read. The 3DM system was able to create a 3D geometry by the direct construction of triangular facets, the extrusion of polylines, and the creation of a set of basic 3D primitives. Due to the important limitation on the resolution of the screens in the available commercial HMDs, other researchers opted for desktop VR solutions such as HoloSketch [3]. This early system used a regular high-resolution monitor (960 × 680, 20 in screen) coupled with stereo shutter glasses to provide stereo vision at a 56.45 Hz refresh rate. It supported a set of primitive 3D objects, free-form tubes, 3D isolated line segments, and polyline wires, among others. The system was a wireframe modeler without any surface or solid representations.

The hardware capabilities, both in terms of processing and visualization, were the limiting factors in the systems developed during these years. Fortunately, the development of low-cost HMDs that began with the DK1 and DK2 Oculus Rift development kits [4] in 2013 and 2014 fueled the development of novel systems that leveraged the affordability of the new generation of HMDs.

The creation of precise geometry in a VR environment has followed two main routes [5]: create a VR front-end to interact with a commercial CAD system or provide 3D modeling functionality using a geometric kernel such as Open Cascade [6] or ACIS [7]. An example of the first approach (linking a VR front-end with a CAD system) is cRea-VR [8], where a middleware architecture was developed to interact with Dassault Systèmes' CATIA V5 in VR [9]. Another prototype system [10] interfaced a CAD system (Autodesk Fusion 360 [11]) with a game engine (Autodesk Stingray [12]) with limited modeling capabilities that supported the creation of solid prisms and spheres, snapping them to a grid, and performing cuts of similar shapes.

Instead of controlling a host CAD system to create 3D geometry, an alternative research line focuses on supporting 3D sketching for VR. For example, Multiplanes [13], provides a VR-based 3D sketching environment that supports 3D freehand drawing that automatically beautifies strokes to compensate for the difficulty of drawing in 3D. Similarly, Smart3DGuides [14] automatically generates visual guidance by analyzing the user's gaze, controller position and orientation, and previous strokes in the VR environment, to increase the overall quality of the drawn shapes. To help users create sketches with higher precision, VRSketchIn [15] combines a 6DoF-tracked pen and a 6DoF-tracked tablet as the input devices, supporting unconstrained 3D mid-air sketching with constrained 2D-surface-based sketching.

Other researchers have used commercial VR applications to study the reaction of users when they have to perform sketching and modeling tasks in the virtual environment. The commercial success of some HMDs has created a market for this type of system. The most widely recognized applications include:

- Gravity Sketch [16];
- Tilt Brush [17];
- Master Piece [18];
- Kodon [19];
- ShapeLab [20];
- Adobe Medium [21];
- flyingshapes [22].

The results are contradictory. For example, a user study [23] with Gravity Sketch and Kodon concluded that these VR applications do not provide significant advances compared to conventional 2D sketching tools. This study also reports that the use of these VR applications produced noticeably more physical fatigue than traditional 2D sketching. Another study [24] compared Gravity Sketch versus paper sketching and offered a positive view by considering VR sketching as a unique form of visual representation that facilitates the rapid and flexible creation of 3D models, engaging users in visual thinking and visual communication activities in ways that cannot be reached with any other tool.

An experiment with Tilt Brush [25] compared paper sketching to VR sketching, with and without a physical surface to rest the stylus. The study concluded that the absence of a physical drawing surface is a major source of inaccuracies in VR sketching. A second experiment analyzed the use of visual guidance to help with the sketching process and improve precision. The authors reported that visual guidance increases positional accuracy, but produces strokes of lesser aesthetic quality.

Another study using Gravity Sketch [26] found that VR modelling is natural and fast and provides better spatial perception for users. However, VR modeling is less precise compared to desktop tools. The study also found that current VR tools do not provide suitable functionality for creating technical objects as they are oriented toward the creation of free-form organic shapes.

*Context and Objectives*

Some of the main learning outcomes of a typical freshmen engineering graphics course are [27,28] 2D and 3D visualization, mapping between 2D and 3D, planar graphical elements, sectional views, methodologies for object representation, projection theory, parallel projection methodologies, drawing conventions, dimensioning, and solid modeling. Unfortunately, many of these topics do not appeal to engineering students [29]. The introduction of new technologies such as augmented reality [30–32] gamification with mobile devices [33] and computer-aided sketching [34] in the area of engineering graphics has always had a positive impact on student motivation. Considering that in the near future, low-cost VR headsets will open a real possibility to add this technology to regular teaching/learning practices in this field, a review of commercial available software [16–22] was conducted to assess their suitability for the type of exercises and topics covered in first-year engineering graphics courses. This review concluded that the learning curve of these systems, their major focus on free-from surfaces, and the difficulty of delivering exercises and tutorials in an integrated manner made them unsuitable for meeting the learning outcomes described previously, leading to the development of the Educational Virtual Modeling (EVM) system described in this paper.

EVM was developed to explore the feasibility of using a simple modeling application to promote students' motivation and support the learning of key topics in the engineering graphics discipline such as modeling and mapping between 2D and 3D.

The option of designing a VR front-end to interact with a commercial CAD system was discarded due to its complexity. Building a VR modeling application on a game engine platform was considered the best alternative. This approach would allow exploring the best way to provide exercises and tutorials to students inside the virtual space and the creation of the classic parts used as exercises in basic engineering graphics courses. These parts are mainly composed of prismatic and cylindrical elements.

In the next sections, the system components will be described in detail to facilitate the understanding of EVM's functionality. Next, the results of a user study with 23 participants are presented to analyze if the design objectives were achieved. Trimble SketchUp [35], a desktop modeling system known for its ease of use, was used as a reference. The paper ends with the discussion of the results and the conclusions, limitations of the study, and future work planned.

## 2. System Description

EVM was developed in C# on the Unity platform Version 2019.2.16f1 [36] and Microsoft's Mixed Reality Toolkit (MRTK) [37], a set of cross-platform tools for the development of mixed reality experiences.

EVM works on a custom database in which all entities can be reduced in their simplest form to a line. For example, a curve is given as a set of consecutive lines, whereas a surface is defined by a contour formed by lines and curves and a set of mesh objects responsible for filling and shading its interior. Figure 1 shows all the surfaces that can be generated with EDM:

- Type 1: when the user drags a line (initial line) along a path (contour curve);
- Type 2: when the user drags the end of a line (initial vertex) along a path (contour curve) while the other end of the line remains fixed in space;
- Type 3: when the user drags a curve (initial curve) along a path (contour curve);
- Type 4: when the user drags the end of a curve (initial vertex) along a path (contour curve) while the other end of the curve remains fixed in space;
- Type 5: by triangulation when the user requests to form a surface from a closed contour. For this type of surface, an ear-trimming algorithm is used [38].

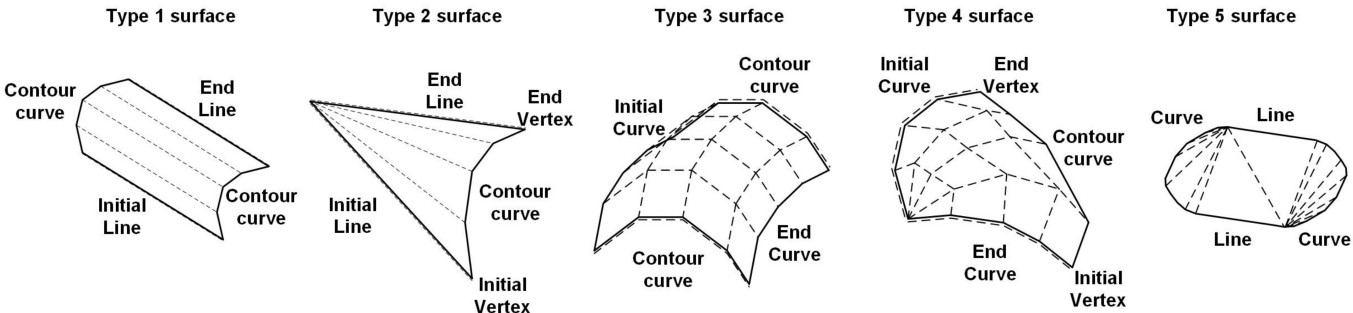

**Figure 1.** Types of surfaces supported by EVM.

For each mesh, a normal vector must be defined to identify the exterior face of the polygon. All this information is necessary to manipulate the model and apply shading and lighting techniques, among other things. In the next section, we discuss the algorithm used to orient the normal vectors to the mesh.

### 2.1. Orientation of Normal Vectors to the Mesh

When the first surface is created in the virtual environment, all the normal vectors in its mesh are oriented in the same direction. The generation of subsequent surfaces triggers a method that reorients the direction of the normal vectors (if necessary) of all meshes of the current surfaces. The goal is to keep all normal vectors pointing toward the outside of a possible solid object.

The algorithm implemented by this function checks each mesh on the surfaces in the scene to determine the number of intersections between a line passing through the center of gravity of a triangle defined by the vertices of the mesh and the mesh of other surfaces in the direction of the normal vector. If the total intersection points is an odd number in any mesh of the surface, the normal vectors of all the meshes in that surface are reversed. An example of how the algorithm works is shown in the process illustrated in Figure 2.

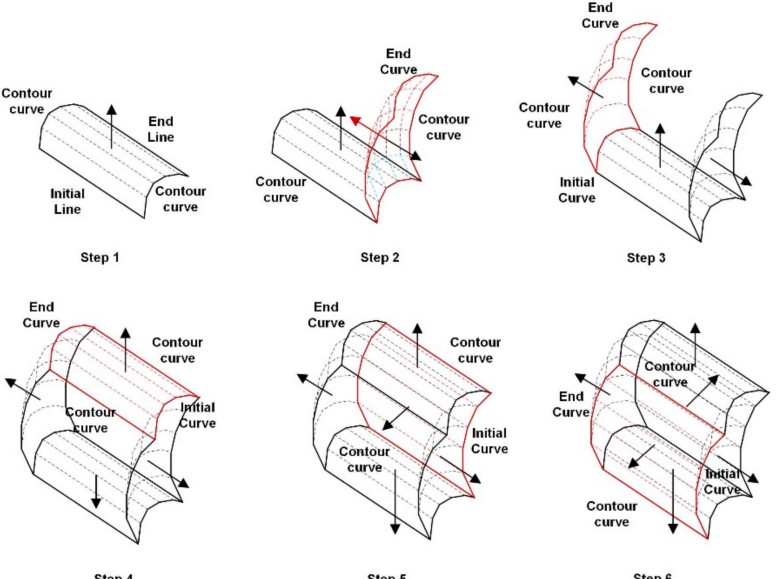

**Figure 2.** Structure of the surfaces represented in the scene.

The sweeping operation can also be performed on surfaces generated in a previous step. In this case, a new surface is produced directly, which is a copy of the original and with the normal vectors in opposite directions. The normal vectors on the first surface are assigned the direction that is opposite the motion of the controller, and those on the second surface are assigned the direction of motion of the controller. Likewise, a new surface is

generated for each entity that defines the contour of the surface being swept. Once again, the normal vectors will point toward the outside of the enclosed volume, as shown in Figure 3.

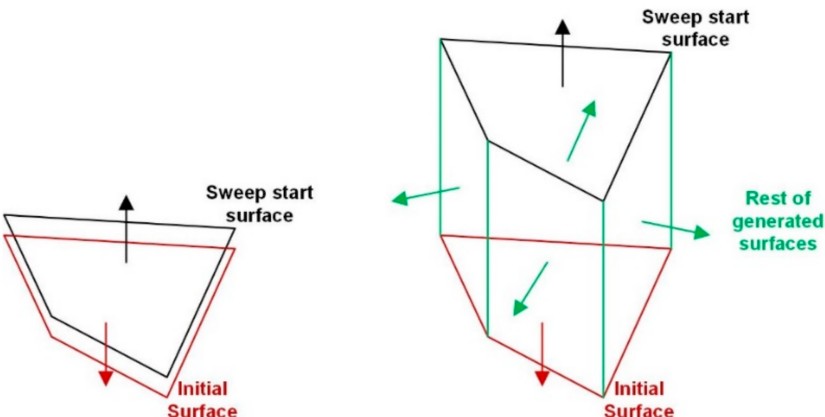

**Figure 3.** Solid generated by extrusion of a surface.

### 2.2. User Workspace

Considering that a main objective of EVM was to provide a simple mechanism for delivering exercises and tutorials to the students in the virtual space, a work area equipped with a main screen for visualizing multimedia tutorial videos and two lateral walls to display the exercises was implemented (model screens in Figure 4).

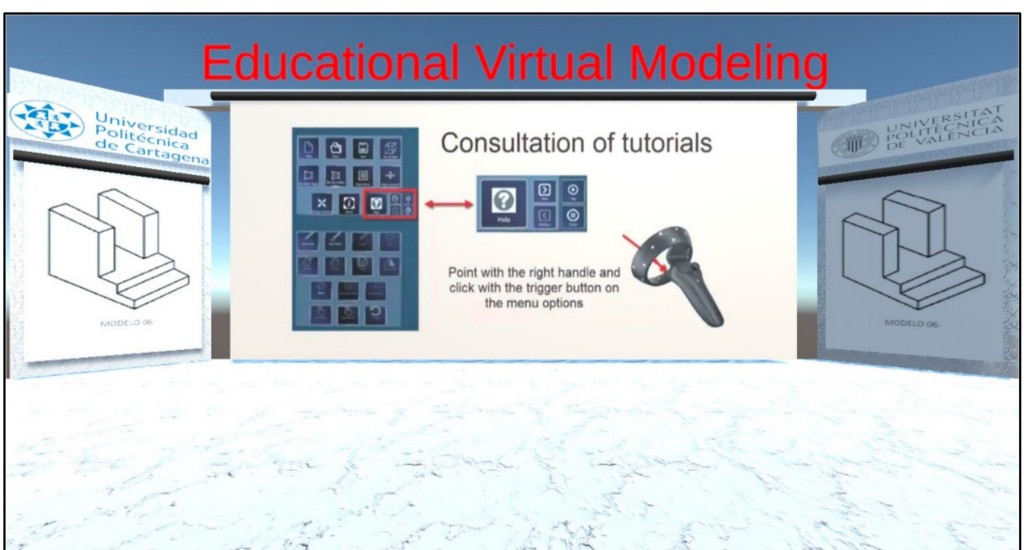

**Figure 4.** User workspace in the virtual environment.

Regarding the user interface, we tested several menu configurations and finally decided to implement a classical panel menu considering our testing results and recent publications [39,40]. The panel menu is attached to a virtual hand that is linked to the non-dominant hand controller. According to the mode of operation, the dominant hand controller can show a pencil (Figure 5 shows an example) for geometry creation and selection, a pencil rotated 180° (with the eraser pointing forward), which indicates the use of the erase command or a virtual hand that is symmetrical to the one shown for the non-dominant hand controller, which indicates the use of a different command. A ray-casting technique is used for the selection.

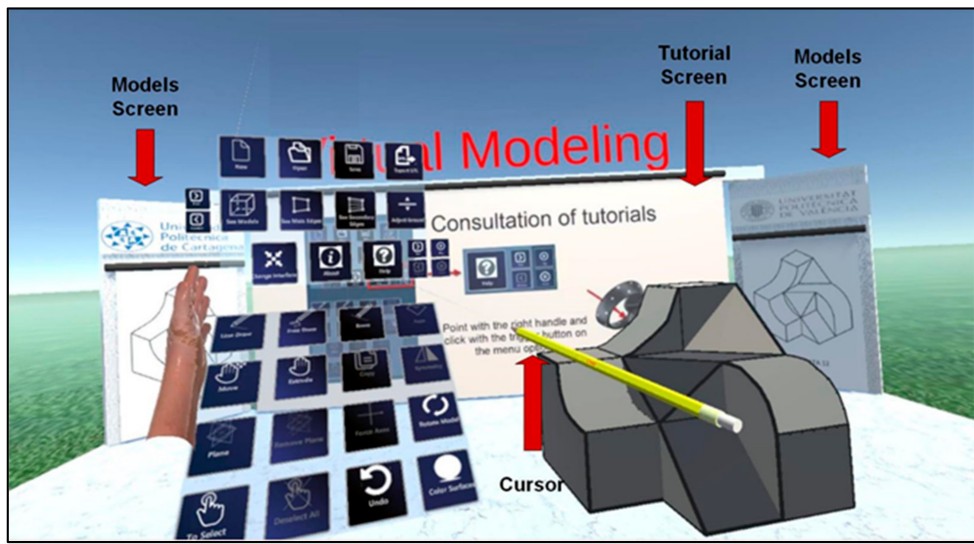

**Figure 5.** User workspace with the panel menu.

Given the educational nature of the application, it is important to highlight the integrated exercise and tutorials' functionality. EVM can visualize exercises stored as image files. The images must be uploaded to a specific directory so they can be visualized using the forward and backward icons from the panel menu. These lateral walls can be hidden by using the icon described in Table 1. The display of tutorials is also one of the strengths of this application. Far from consulting manuals or information in any other medium, the application can display video tutorials on the central wall. For this purpose, a total of four tutorial videos with a duration of around 9 min were recorded. The user can watch these videos by clicking the play, pause, next, and previous icons deployed when selecting the help icon shown in Table 1.

**Table 1.** Commands to manage tutorials and lateral walls.

| Icon | Description | Icon | Description |
|:---:|:---:|:---:|:---:|
| Show Models | Shows/hides the lateral walls with the images of the exercises to be performed. | Help | Shows/hides the help menu, based on video tutorials. |

### 2.3. Commands

The panel menu presents a set of icons organized into four groups: commands to create geometry, commands to edit geometry, modeling aids, and commands to manage files, visualization, and settings.

The commands available to create geometry in the virtual environment are shown in Table 2. During the execution of any command, surfaces, endpoints, and midpoints are detected automatically on any line or curve in the virtual space. These points can be selected as start or end points for the command.

The extrude command allows users to sweep previously drawn lines or curves. The user can click and drag an existing entity or perform the action on the initial or final vertices, generating some of the surface types described at the beginning of Section 2. Additionally, as described in Figure 3, it is possible to sweep surfaces to create volumes. The path followed by the geometry during the sweeping operation follows the trajectory of the dominant hand controller. The commands face, copy, and symmetry are used on a set of pre-selected entities. For the face command, the selected entities must be coplanar and define a closed contour.

**Table 2.** Available commands to create geometry.

| Icon | Description | Icon | Description |
|---|---|---|---|
| Line Draw | Draws a line in the virtual space. | Extrude | Sweeps a line, curve, or surface along a straight path for surface creation |
| Free Draw | Draws a curve in the virtual space. | Copy | Creates a copy of the selected geometry. |
| Face | Creates a flat face from selected lines or curves. | Symmetry | Creates an object symmetrical to the one selected about a fixed plane. |

The commands available to edit the existing geometry in the virtual environment are shown in Table 3. Moving the location of an entity implies that any geometry that the entity belongs to must be modified. In fact, if a vertex changes its location, any line, curve, or surface that connects to that point will readjust automatically to the new location of the vertex. If a line or curve is moved, then any surface that contains the line will readjust to the new location of the line or curve.

**Table 3.** Available commands to edit geometry.

| Icon | Description | Icon | Description |
|---|---|---|---|
| Move | Changes the location of an object in the virtual space. | Erase | Deletes an existing entity. |

To facilitate the creation of specific geometric shapes, we implemented a series of modeling aids to the proposed system, as shown in Table 4. The "plane" command is used to create work planes. Once a plane is defined, all commands for creating geometry will be executed with respect to this plane. The "delete plane" command deletes any previously created plane. The "force axes" command can only be activated when there is an existing plane in the scene. The command forces all geometry creation and editing operations to be performed along the main axes that define the plane. Selecting geometry involves clicking on it. Selecting an entity that has previously been selected will deselect it. The "deselect" command will clear any selections in the scene. Finally, the undo command erases the last change made to the model, reverting it to an older state.

**Table 4.** Modeling aids.

| Icon | Description | Icon | Description |
|---|---|---|---|
| Plane | Creates horizontal, frontal, or profile planes, or planes defined by three points. | To Select | Selects an entity in the virtual space. |
| Delete Plane | Deletes an existing work plane in the virtual environment | Deselect All | Deselects a previously selected entity in the virtual space. |
| Force Axes | Forces operations along the 3 axes of the work plane. | Undo | Undoes the last modeling operation |

The commands available to manage files, visualization, and settings in the virtual environment are shown in Table 5. These commands enable the creation of new files, as well as opening and saving a 3D model. We also included the option to export the model to STL format, so that the model can be viewed in or printed from other applications. It also confirms the validity of the geometric information created by our modeling tool and stored in our database.

**Table 5.** Commands to manage files, visualization, and settings.

| Icon | Description | Icon | Description |
|:---:|:---:|:---:|:---:|
| New | Deletes the current virtual scene and creates a new one. | Color Surfaces | Changes the color of the selected surfaces. |
| Open | Open an existing model in the virtual space. | Rotate Model | Rotates the model. |
| Save | Saves the current model to a file. | Adjust Ground | Adjusts the ground level of the virtual space. |
| Export STL | Exports the model to binary STL format. | Show Models | Shows/hides the roll up screens with the images of the objects to be modeled. |
| Show Main Edges | Shows/hides the edges that define the contour of a surface. | Help | Shows/hides the help menu, based on video tutorials. |
| Show Secondary Edges | Shows/hides the edges that define the mesh of a surface. | About | Shows/hides the credits. |

The commands "show main edges" and "show secondary edges" allow users to show/hide the edges that define the contour or the mesh of a surface, respectively. A 3D model with secondary edges visible, a model with some colored surfaces (using the color surfaces command), and a model visualized after being exported to STL are shown in Figure 6.

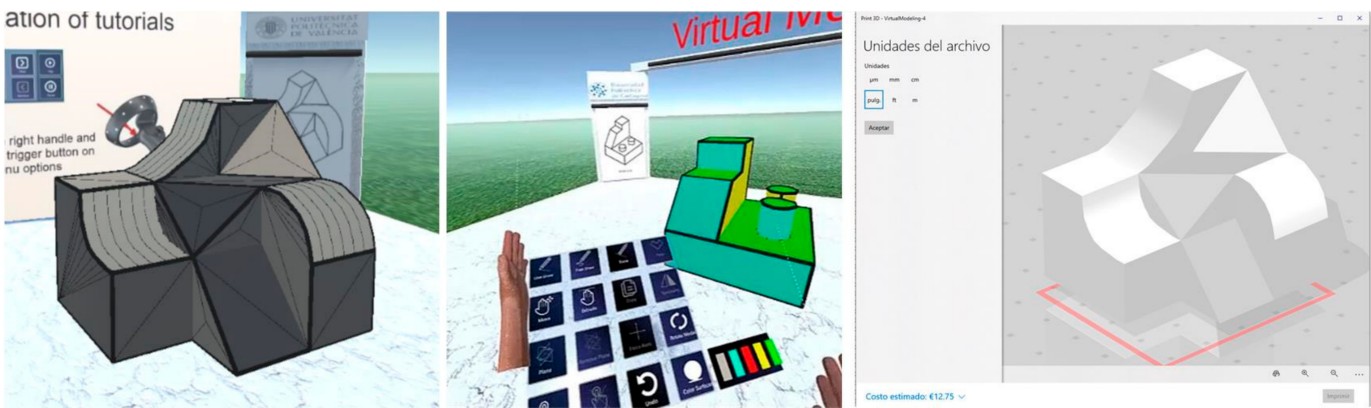

**Figure 6.** Model with secondary edges visible (**left**), surface coloring (**middle**), and exporting a model to STL (**right**).

## 3. User Study

To determine if EVM provides a simple modeling mechanism to support the creation of typical parts used in basic engineering, a user study was conducted with 23 participants. We used Trimble SketchUp as a reference to compare the performance of EVM. SketchUp is a desktop CAD system used in many introductory engineering and architecture graphics courses. It is well known for its simplicity and ease of use. SketchUp follows a modeling paradigm similar to EVM. 3D models are created by defining 2D contours that can be swept to generate volumes. These volumes can then be modified by manipulating their vertices, edges, and faces.

### 3.1. Participants

Twenty-three first- and second-year industrial engineering students (five females and eighteen males) between the ages of 18 y and 21 y and with no previous experience in any of the systems tested were selected to participate in our study.

### 3.2. Materials and Instruments

Both EVM and Trimble SketchUp ran on an HP computer with an Intel Core I7 2.2 GHz processor with 8 GB of RAM and a 64 bit Windows operating system. For the virtual reality application, we used a Lenovo Explorer HMD with a 1440 × 1440 resolution per eye and 90 Hz refresh rate and two motion controllers. The study was set up in a room with an available space of 4 m². The participants had to perform 6 modeling tasks, using the axonometric drawings shown in Figure 7 as the input. The sequence of modeling tasks was generated random for each participant. The parts were not dimensioned as the subjects were asked to model 3D objects with proportions as those observed in the axonometric perspectives.

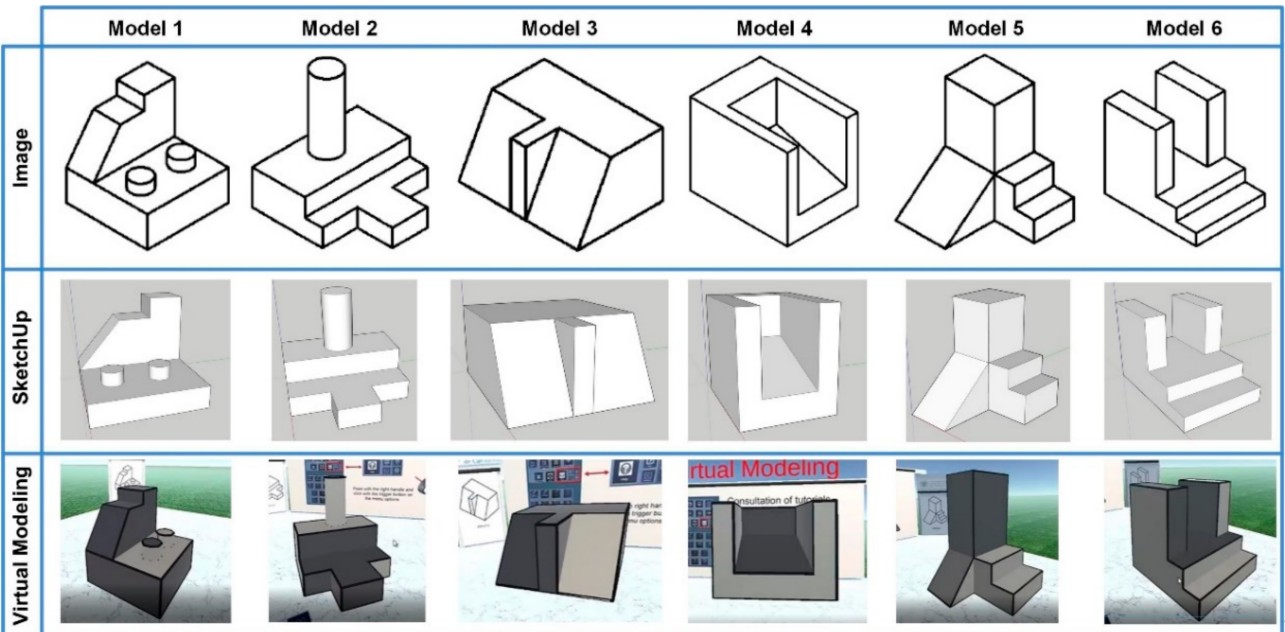

**Figure 7.** Set of models used in our experiment.

The System Usability Scale (SUS) [41] questionnaire (presented on Table 6) was used for the evaluation of perceived usability. Although published in 1996, it is still widely used in many user studies [42], especially in the areas of 3D modeling and VR [15,43–45]. The instrument uses 5-point Likert scales (1 indicates that the user strongly disagrees with the statement and 5 that he/she strongly agrees).

**Table 6.** System Usability Scale (SUS) questionnaire [41].

| Q1 | Q2 | Q3 | Q4 | Q5 |
|---|---|---|---|---|
| I would use this tool frequently | I find this tool unnecessarily complex | The tool was easy to use | I would need the help of somebody with technical knowledge of this tool | The tool's functionality is well integrated |
| **Q6** | **Q7** | **Q8** | **Q9** | **Q10** |
| The tool is inconsistent | Most people could learn how to use this tool very quickly | The tool is very difficult to use | I feel confident using this tool | I had to learn many things before I could use this tool |

For the assessment of the ease of use, we applied the same questionnaire (shown in Table 7) that Stadler et al. [44] used to evaluate ImPro, a new application for immersive prototyping for industrial designers with a similar structure to EVM. It also employs a 5-point Likert scale as the SUS questionnaire.

**Table 7.** Ease of use questionnaire [44].

| Q1 | Q2 | Q3 | Q4 | Q5 |
|---|---|---|---|---|
| Navigating the command menus was not a problem for me. | Navigating the scene was not a problem for me | Changing the scale of the scene was not a problem for me | Finding the command I was looking for was not a problem for me | Creating a 3D primitive was not a problem for me |
| **Q6** | **Q7** | **Q8** | **Q9** | **Q10** |
| Drawing in 3D was not a problem for me | Deleting an object was not a problem for me | Editing an object was not a problem for me | Copying an object was not a problem for me | Undoing and redoing the last command was not a problem for me |

### 3.3. Experimental Procedure

Separate appointments were made for each participant in the study. At the beginning of the experiment, each participant went through a brief training session with each tool where he/she watched a tutorial and was allowed to use the tool freely for a period of 20 min. Subsequently, participants were asked to model each of the proposed parts (the time required to complete the modeling task was registered). The order was randomly chosen for each participant. To minimize bias, 50% of the participants started the experiment using SketchUp and the other 50% using EVM. At the end of the modeling tasks, the participants filled out the SUS and ease of use questionnaires.

### 4. Results

The average modeling times employed for each participant in each of the parts in our study are shown in Figure 8. Except for Models 3 and 6, the time required to create a model using EVM was greater than the time required to perform the same task using SketchUp. For Model 6, both tools performed similarly. The boxplots corresponding to modeling each part are illustrated in Figure 8. The execution time for each participant is presented in Figure 9. Boxplots for execution times are displayed in Table 8.

After confirming the normality of the data with a Shapiro–Wilk test and applying a repeated measures $t$-test with Bonferroni correction to compare the mean execution time for each model, we observed statistically significant differences between EVM and Trimble SketchUp except for Model 6, as detailed in Table 8.

Regarding the SUS score results, the mean value of the SUS score using EVM was 92.3 points, with a standard deviation of 6.15, and 76.30 points for SketchUp, with a standard deviation of 6.69. A repeated measures $t$-test concluded that there existed a statistically significant difference between the SUS scores, $t(22) = 8.89$, $p < 0.001$, which confirmed the observation of the boxplot in Figure 10.

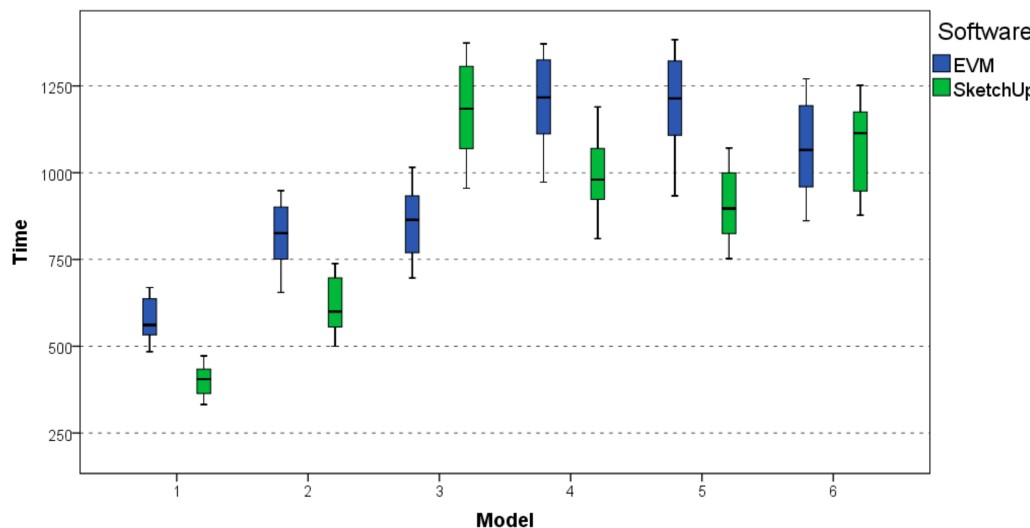

**Figure 8.** Boxplot for each modeling task using EVM and Trimble SketchUp.

**Figure 9.** Times employed by the participants to create the six models using SketchUp and EVM.

**Table 8.** Mean execution time (in minutes) per model and total, std. deviation, and *t*-test results.

| Model | EVM | Trimble SketchUp | Repeated Measures *t*-Test |
|---|---|---|---|
| 1 | 9.67 (1.04) | 6.55 (0.70) | $t$ (22) = 14.87 $p < 0.001$ |
| 2 | 13.63 (1.46) | 10.40 (1.33) | $t$ (22) = 7.31 $p < 0.001$ |
| 3 | 14.27 (1.73) | 19.72 (2.25) | $t$ (22) = −8.78 $p < 0.001$ |
| 4 | 20.12 (2.19) | 16.43 (1.68) | $t$ (22) = 7.30 $p < 0.001$ |
| 5 | 20.03 (2.24) | 15.17 (1.77) | $t$ (22) = 9.08 $p < 0.001$ |
| 6 | 17.73 (2.19) | 17.90 (1.92) | $t$ (22) = −0.31 $p = 0.76$ |
| Total | 95.44 (5.81) | 86.27 (2.51) | |

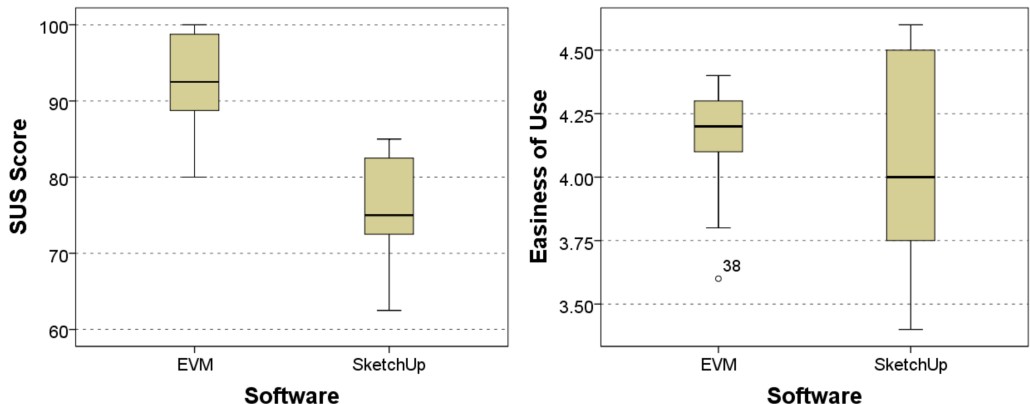

**Figure 10.** Boxplots for SUS and ease of use scores.

Considering the ease of use score, the mean value for the SUS score using EVM was 4.17 points, with a standard deviation of 0.19 and a median of 4.2. SketchUp obtained a mean value of 4.06 with a standard deviation of 0.43 and a median of 4. Because of the non-normality of the data, we applied a related-samples Wilcoxon signed rank test, which revealed that there were no statistically significant differences between the ease of use scores (Z = −1.32, $p = 0.18$). The corresponding boxplot is presented in Figure 10.

The relative SUS and ease of use scores for each participant (the ease of use score was normalized to a 100-point scale) are presented in Figure 11. The relative score was calculated by subtracting the SketchUp score from the EVM score. Positive values indicate better scores for EVM, and negative values refer to SketchUp. The rectangles represent the score distributions between first and third quartiles. The horizontal line inside the box represents the average score. The relative scores per question are shown in Figure 12.

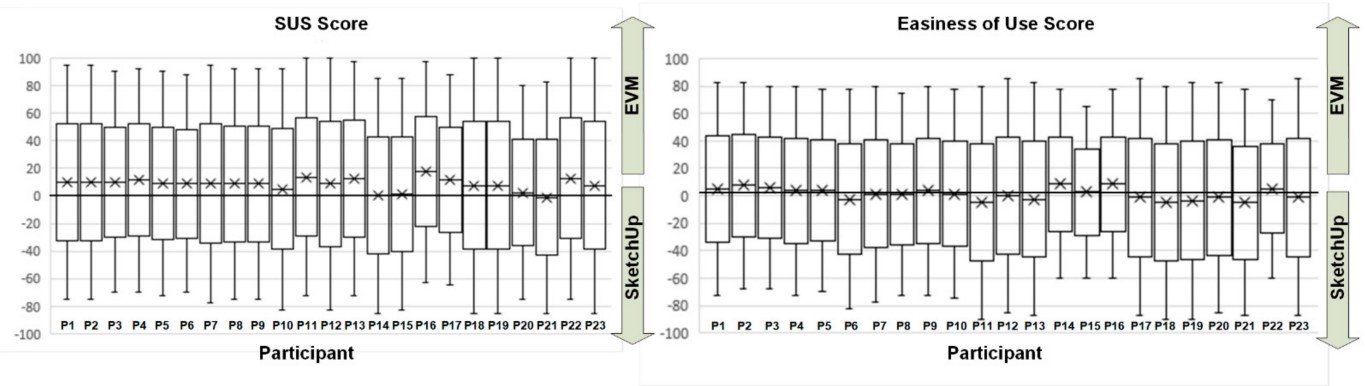

**Figure 11.** Relative SUS and ease of use scores per participant.

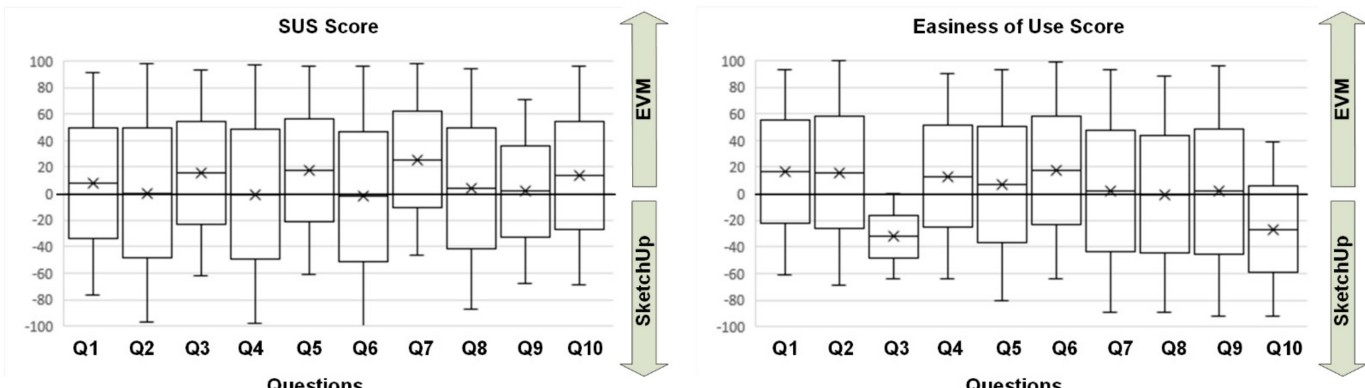

**Figure 12.** Relative SUS and ease of use scores per question.

Figure 11 shows that all participants rated EVM higher than SketchUp in the SUS questionnaire except for one participant. In the ease of use questionnaire, 9 participants scored SketchUp higher, 1 participant scored both applications equally, and 13 considered EVM to be easier to use.

Analyzing the questionnaires per question, EVM always obtained more positive scores for the SUS than SketchUp except on Questions Q4 and Q6 (by a small margin). In terms of ease of use, EVM received higher scores except on Questions Q3 and Q10, which were related to the scaling operations and undo/redo operations, which makes sense as EVM does not provide a scale or redo command.

## 5. Discussion

The observations of the participants during the experiment and their comments during the informal interview at the end of the experiment provided interesting information to interpret the results. The initial reaction of participants when they tried a VR HMD for the first time was astonishment and enthusiasm. All of them seemed much more excited to work in VR than with the desktop program. The 20 min they were allowed to practice freely were short when they did this with EVM, and they wanted to continue modeling and experimenting. However, with SketchUp, the 20 min were very long, and they wanted to move on to the next experimental phase sooner. This is a relevant fact to consider EVM as a potential learning tool in an academic setting, as learner motivation is an important part of any learning experience. If learners are not motivated, no matter how effective a particular tool is, they will not use it. This observation confirms the results of previous works [46–48] and reflects the positive impact of VR in increasing student motivation.

Regarding the execution time of the modeling tasks, the observation of the behavior of the participants may explain why SketchUp outperformed EVM in four of the six modeling tasks. We highlight two key aspects:

1.  When modeling in the VR-based environment, participants tended to create models that were much larger in size than those created in SketchUp, which forced users to move around the virtual environment and perform actions that required the movement of their arms. Working with SketchUp, however, only requires subtle movements of the hands and wrists, which undoubtedly can be performed much more quickly. This behavior is similar to other studies [23] that reported physical fatigue in the VR condition due to the physical activity required to sketch and model in that context;

2.  When modeling in the VR system, participants tended to spend a significant amount of time looking at the model and the environment. This behavior is common in users who have never experienced an immersive virtual reality environment before. This behavior perhaps is related to observations in similar experiments [24] that found a simpler mental image manipulation as one of the positive characteristics of VR sketching. The participant does need to create a 3D mental representation before sketching, as a virtual 3D representation supports cognitive processes.

It is important to mention the important difference in the perceived usability measured by the SUS score. Considering the extensive analysis performed in [49], the high score (92.3) obtained by EVM is very relevant and can be rated as excellent. The SketchUp score was modest (74.3), which can be rated as good. These results can be partially explained by considering the typical "wow effect" [50] that first-time VR users experience.

Regarding the ease of use, the similar scores of EVM and SketchUp and the fact that no statistically significant differences were found in the questionnaire's overall score may be explained by the fact that the ray-casting and panel menu combination used in EVM are conceptually very near the typical WIMP interface [51], which is well known by participants. Mouse cursor operation and selection of menu options by clicking the mouse on an icon are replaced by moving the controller in space and using ray-casting to select the desired icon.

## 6. Conclusions, Limitations, and Future Work

After the analysis of the results of the user study, we can conclude that EVM can be a feasible tool for an introductory engineering graphics course. The positive impact on motivation, high score on the SUS questionnaire, equivalent ease of use as the SketchUp system (well known for its simplicity), and task completion times in a range near (95.4 min for EVM vs. 86.27 for SketchUp, as the total modeling time) SketchUp support this statement. It is important to mention that EVM was not designed to provide a faster way to model. The goal is to create a tool that is viable to deploy in a laboratory and provides reasonable task completion times. After the experiment was completed, conversations with participants revealed that the experiential learning facilitated by VR had a greater impact on students than a desktop application. Taking this into account, the small difference in total execution time is irrelevant.

A significant contribution of the developed system is the immersive help system. One of the most praised features of EVM was the possibility of viewing multimedia tutorials on how to use the application within the virtual environment itself, as well as using the side virtual screens to view the modeling tasks that had to be performed. In addition, the fast learning curve and ease of use shed light on the feasibility of using a virtual reality environment to perform basic sketching and modeling tasks with minimal training. These features, coupled with the fact that HMDs are becoming less expensive and more powerful, justify investing the additional effort to improve the developed system.

As future work, we plan to improve the capabilities of EVM by first porting the application to a standalone HMD such as Oculus Quest 2 and then completing the functionality of the application with elements that the user study revealed, such as the possibility of scaling parts and a complete implementation of undo/redo commands. Other ideas that are being considered to enrich the current system to facilitate the learning of spatial relations is the representation of the lateral projections of the objects under construction, to be able to easily analyze the connection between 3D objects and their orthogonal projections, which are often a difficult concept to work with in traditional teaching environments. Finally, we would like to conduct user studies with larger sample sizes and longer interventions to overcome the limitations of the study presented in this paper, which had a sample size of 23 subjects and a duration of the modeling tasks of approximately 90 min.

**Author Contributions:** Conceptualization, J.C.-P. and M.C.; methodology, M.C.; software, J.C.-P.; validation, J.C.-P.; writing—original draft reparation, J.C.-P. and M.C. All authors have read and agreed to the published version of the manuscript.

**Funding:** This research received no external funding.

**Institutional Review Board Statement:** Not applicable.

**Informed Consent Statement:** Informed consent was obtained from all subjects involved in the study.

**Data Availability Statement:** The data presented in this study are available on request from the corresponding author.

**Conflicts of Interest:** The authors declare no conflict of interest.

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
