# Peer review of "EVM: An Educational Virtual Reality Modeling Tool; Evaluation Study with Freshman Engineering Students"

_applsci, doi:10.3390/app12010390_

Round 1

Reviewer 1 Report

TITLE

Virtual Modeling: A Virtual Reality Modeling Tool oriented to Mechanical Part Ideation.

SUMMARY OF CONTRIBUTION

The paper describes the definition, implementation and testing of a VR-based tool for generating 3D models.

FINAL EVALUATION

The paper does not match at all the expectations coming from reading the title and the abstract. The tool proposed cannot be considered a real aid to “ideate” mechanical parts. It is a tool to generate simple shapes (without real dimensions) for students or newbies in general. Moreover, concepts behind the development of this tool are 30-year old (except for the VR aspects, of course); the UX evaluation is full of misunderstandings such that the results cannot be considered as meaningful… In summary, the paper, in its current format, should be rejected. A change of perspective, moving from a tool for mechanical part ideation to an educational one, along with a deep review of the validation section, could give the paper some chances but, at that point, I do not know how much it would fit with the Applied Science Journal specificities.

COMMENTS

Although I suggest to reject the paper, I add some comments/suggestions for the authors to help them improving their research and the quality of the papers that will come from it.

TITLE

As said before, the research (and the papers from it) could have a chance if it (and them) will be named and addressed properly. I mean, here we have an educational tool to make people get in touch with 3D models, tridimensional working spaces, etc., all of this with the aid of the VR. What we HAVE NOT here, is a “modeling tool oriented to mechanical part ideation”. If this were clear in the paper title, the reader would approach it differently, more effectively, more positively I guess.

“Virtual Modeling” is absolutely too generic. The authors should assign a different name to their system, more meaningful, more recognizable.

ABSTRACT

The abstract does not make clear that Virtual Modeling is the system developed by the authors.

“The participants filled out a usability test (SUS questionnaire)”… here my first doubts on the authors’ knowledge about UX matters started raising. “to fill a usability test” sounds quite strange as soon as you know the fundamentals of UX. Other clues about the authors’ scarce confidence with UX came afterwards.

The last paragraphs show some incongruences. If, on one hand, the authors tell that the desktop application performed better, on the other hand, they reward the VR-based application too. Unfortunately, the way they assign credits to the latter makes it appear even worse (it made people lose time looking around, move around because the models were too big, get tired, etc.). The same problem shows up in the discussion section of the paper.

  1. INTRODUCTION AND RELATED WORK

The content of this section is ok. The context, goals and document structure subsections are correctly placed. Nevertheless, to me, the first 15-20 lines are useless. They describe very old, too old concepts.

The kind of tools described at lines 93-97 seems to match exactly that of the tool proposed in the paper (Virtual Modeling). If this is true, it should be made explicit.

  1. SYSTEM ARCHITECTURE

Figure 1 is straightforward, then, useless. I cannot call it an “architecture”. They are just three pictures connected by three arrows.

Figure 3 is quite messy. There are redundancies in it; moreover, the classification (ruled vs. curved surfaces) is not highlighted properly.

The description of the data structure and functioning of the Virtual Modeling is too long. It goes deep into specificities that do not need to be described in this kind of paper. For example, subsection 2.3 (orientation of normal vectors in mesh) makes no sense here. All of this steals space to much more important matters like the UX evaluation, the comparison between the Virtual Modeling and the desktop app, etc.

  1. SYSTEM FUNCTIONALITY

“The interface is the main menu…” this is the second clue that made me think about big gaps between the authors and UX matters. I think that there is no need to go further in explaining why.

Unless I missed it, there is no way to assign precise dimensions to the models generated thanks to the VR-based system. This is enough to make considering it as a “Mechanical Part Ideation tool”… just exotic.

  1. USER STUDY

Reading this section, I had the most of the doubts, questions, perplexities, etc.

Six models were asked to be created using Virtual Modeling and the desktop app. That’s ok. But… in which order? Which first? This is extremely important (but the authors do not address this) because it could have a deep impact on the evaluation results in terms of bias.

Where are the criteria leading the model selection?

Which were the requirements (precision, modeling speed, etc.)?

“the participants filled out a usability test and a functionality test”… Usually, questionnaires, forms, etc., are filled; tests are performed.

“the usability of the user experience”… ok. This is the ultimate clue about the authors’ missing knowledge about UX. I suggest to read some fundamentals about UX before to go further. UX fundamentals will make clear that UX (and CX) are the state of the art, that usability is 30-year old, that you cannot deal with usability and functionality separately (which cause the problems to which?), that you cannot neglect the bias in evaluating the UX, etc.

SUS is very old (1986). There are much more recent methods and tools to address UX evaluation, methods and tools able to consider all the aspects listed before. Among them, for example, I suggest to give the meCUE questionnaire a try.

All this said, it is straightforward that the results of the comparison between the Virtual Modeling and the desktop app as they appear in the paper, from both the usability and the functionality points of view, are unreliable, meaningless.

Table 6 contains ten questions. All of them are formulated in the negative tense. This should be avoided in any serious questionnaire.

Figure 18 is badly positioned. It should appear before lines 409-411.

The sentence of lines 409-411 misses the subject (“Virtual Modeling”).

  1. DISCUSSION AND CONCLUSIONS

“Despite the seemingly lower performance”, “we should avoid negative interpretations”. There is something wrong in this. Simply… the former is objective (“lower performance”), the latter is subjective (“we should avoid…”). We are comparing apples with pears.

As said at the beginning, the way the authors try to give credits to the Virtual Modeling make things even worse. For example, the “wow effect”, in this case, is something negative. If the VR involvement makes the users perform worse, the wow effect makes things even worse!

“Our results demonstrate the value of the Virtual Modeling” I do not agree at all (not to mention that, as said before, without any knowledge about UX evaluation methods and tools, the results shown in the paper are meaningless).

THROUGHOUT THE PAPER

There are some misspellings here and there. In case of positive evaluation of the paper, a double check of the English language would have been suggested.

Reviewer 2 Report

Though the study has some original contributions, it needs major revisions before publishing as follows:

  • The research gap(s) that the study has tackled needs to clearly be mentioned (My suggestion: develop them in the form of research questions at the end of Introduction section).
  • The authors need to clearly state the contributions made to the body of relevant knowledge in Conclusion section (use numbers such as 1,2,…).
  • The authors should state the significance of the study.
  • Discussion section should be segregated from Conclusion section.
  • The authors need to add limitation and potential future research to the manuscript.

Reviewer 3 Report

The authors present a new VR tool development and Its user study evaluation to compare its functionality and usability. Please improve your explanation of the objectives of the paper  (see page 3 lines 102-114). Is the development of this new tool or the user study evaluation to compare its functionality and usability?

Some references are missing, please review section 1  

See page 2 lines 21-27 and line 113, the authors could Include these links as references in the reference section. 

The development presented does appear to have some merit, but the paper would have to be re-written in a more concise way (see sections 2 and 3). There is also a lot of information that could distract the reader.  It is interesting to read it, but it is not necessary to share too many of these. These sections would have to be re-written in a more concise way.

The manuscript must describe a technically sound piece of scientific research with data that supports the conclusions. Experiments must have been conducted rigorously, with appropriate controls, replication, and sample sizes. The conclusions must be drawn appropriately based on the data presented. The authors did not provide a complete statistical analyses procedure. Is there any relationship between the questions and, for example, the samples?   A new statistical analysis should be added to the paper, not only a descriptive statistical analysis.

Limitations of the results of the evaluation could be included in the section of Discussion and results.

Round 2

Reviewer 2 Report

Following the improvements made to the manuscript, it can be considered for publishing.

Author Response

Thanks for your final decision

Reviewer 3 Report

The authors have responded to most of this reviewer's suggestions.

In this reviewer's opinion, the title could be more accurate if the authors focused it on the evaluation of the academic tool they propose. And  if the authors focus the title only on the evaluation of their academic tool, the manuscript is more appropriate. This idea can be read in the new section 1.2. Context and objectives.

Author Response

Following reviewer indication a new title has been chosen:

EVM: An Educational Virtual Reality Modeling Tool. Evaluation Study with Freshman Engineering Students